



# A new reference-quality precipitation gauge wind shield

John Kochendorfer[1], Tilden P. Meyers[1], Mark E. Hall[1], Scott D. Landolt[2], Howard J. Diamond[3]

[1]Atmospheric Turbulence and Diffusion Division of the Air Resources Laboratory, National Oceanic and Atmospheric Association, Oak Ridge, TN, 37838, USA

[2]National Center for Atmospheric Research, Boulder, USA

[3]Atmospheric Sciences and Modeling Division of the Air Resources Laboratory, National Oceanic and Atmospheric Association, College Park, MD, 20740, USA

*Correspondence to*: John Kochendorfer (john.kochendorfer@noaa.gov)

**Abstract.**

Gauge-based precipitation measurements suffer from undercatch due to the effects of wind, with solid precipitation measurements especially susceptible to such errors. When it is snowing and windy, unshielded precipitation gauges can catch less than half of the amount of precipitation of a gauge that is protected from the wind. For this reason, the US Climate Reference Network (USCRN) developed a large, double layer, wooden wind shield called the Small Double Fence

Intercomparison Reference (SDFIR). In past studies, the SDFIR has been demonstrated to be the most effective wind shield in use in any weather or climate network, reducing solid precipitation undercatch to less than 10% in wind speeds up to 8 m s-1. However, the wooden SDFIRs are subject to decay, they are difficult to replace and maintain, and they hinder access to maintaining the gauge. For these reasons, a new precipitation gauge wind shield called the Low Porosity Double Fence (LPDF) has been developed for use in the USCRN. Tested at three separate sites chosen for prevalent windy and snowy weather, the

precipitation measurements recorded within the LPDF compared well to the SDFIR. After more than two years of measurements, the total precipitation recorded by the LPDF at each individual site differed by ± 1.2%, and the total LPDF accumulation from all sites was 0.03% greater than the SDFIR accumulation. For the measurement of solid precipitation, the LPDF-shielded measurements were statistically indistinguishable from the SDFIR, and the time series of accumulation from precipitation gauges shielded by the SDFIR and the LPDF were almost identical. This new wind shield is much smaller and

easier to install and maintain than any other reference-quality wind shield for the measurement of solid precipitation, and may be of use within other meteorological, hydrological, and climate networks. It could also serve as a secondary reference precipitation measurement for precipitation intercomparisons held in remote locations where the construction of a full-sized Double Fence Intercomparison Reference (DFIR) shield is not feasible.

**1 Introduction**



No weather phenomenon is as destructive or yet as essential in all seasons as precipitation; it causes floods, avalanches, and landslides; and conversely a lack of precipitation can lead to devastating droughts while also supplying the source for

drinking water and growing food. Since all terrestrial life relies on it, humans have been developing methods to best quantify, understand, and predict precipitation for thousands of years. Accurate precipitation measurements are required for the development and improvement of modern climate, weather, and hydrologic models (e.g. Buisán et al., 2020; Køltzow et al., 2020; Larson and Peck, 1974; Widmann and Bretherton, 2000; Tapiador et al., 2017; Rozante et al., 2010). Reference-quality precipitation measurements are required in order to calibrate and validate precipitation products such as remotely sensed and

gridded precipitation measurements (Chen et al., 2008; Rajulapati et al., 2020; Adam and Lettenmaier, 2003; Henn et al., 2018; Kluver et al., 2016; Newman et al., 2015; Shi et al., 2017; Larson and Peck, 1974; Poméon et al., 2017). The detection and monitoring of precipitation trends associated with climate change also require reliable and bias-free precipitation measurements. Because the phase of precipitation is predicted to change in some regions (e.g. Trenberth et al., 2003; Trenberth, 2011; Prein and Heymsfield, 2020; Coats, 2010), measurement errors that are affected by precipitation phase may be especially

difficult to disentangle from changes in precipitation amount as Earth's climate warms. This makes it especially important to record accurate, climate-quality measurements of precipitation amount, irrespective of whether the precipitation occurs as rain (liquid precipitation), mixed phase, or snow (solid precipitation).

Solid precipitation measurements are subject to significant undercatch caused by wind (Goodison, 1978; Golubev, 1986; Sevruk et al., 1991; Groisman and Legates, 1994; Yang et al., 1995; Goodison et al., 1998; Yang et al., 1999; Macdonald and

Pomeroy, 2007; Smith, 2009; Nitu et al., 2019; Cauteruccio et al., 2021; Leroux et al., 2021; Thériault et al., 2021). Because pit gauges cannot be used in areas where it snows, wind shields are used to help reduce the wind speed around the gauge and improve the accuracy of precipitation measurements (e.g. Alter, 1937; Groisman et al., 1991; Nitu et al., 2019; Baghapour and Sullivan, 2017; Colli et al., 2016; Wolff et al., 2015; Goodison, 1978). The World Meteorological Organization Solid Precipitation Intercomparison Experiment (WMO-SPICE) established a new reference for the automated measurement of solid

precipitation, which includes a large, three-layered wind shield around the gauge called the Double Fence Automated Reference (DFAR). The DFAR is essentially comprised of the same two outer fences as the Double Fence Intercomparison Reference (DFIR) shield originally designed for manual solid precipitation reference measurements, with a single Alter shield and an automated weighing gauge at its centre (Ryu et al., 2012; Nitu, 2012; Rasmussen et al., 2012; Nitu et al., 2019). Testing performed as part of WMO-SPICE also demonstrated that wind shielding is the most significant determinant of solid

precipitation errors, with different weighing gauges performing similarly when they were similarly shielded (or unshielded) (Kochendorfer et al., 2018; Nitu et al., 2019). This testing included gauges shielded within the US Climate Reference Network (USCRN) Small DFIR (SDFIR), the Belfort double Alter shield, the double Alter shield, the single Alter shield, and unshielded gauges. In addition to establishing and quantifying the importance of shielding, this work indicates that the porosity of the wind shield plays an important role in determining its efficacy. Based on wind break and turbulence research (Wilson, 1987;

Hagen and Skidmore, 1971; Heisler and Dewalle, 1988; Středová et al., 2012; Wilson, 1985), Belfort designed a new double





Alter shield with a porosity of 25%, which is in contrast to the 50% porosity of most other wind shields. Due to its lowered porosity, the Belfort double Alter was found to be almost as effective as the much larger SDFIR (Kochendorfer et al., 2017b).

Due to the size (12 m diameter and 3 m tall) of the Double Fence Intercomparison Reference (DFIR) wind shield and the amount of material and labour required to construct and maintain it, this shield is not appropriate for use in operational networks. For this reason, a smaller version of the DFIR, called the Small DFIR (SDFIR) is used by the USCRN at the 114 sites throughout the contiguous US. Like the DFIR shield, the outer two shields of the SDIFR are made of wood, and require regular maintenance. The USCRN was established over 20 years ago (Diamond et al 2013), and many of the SDFIRs within the network now need to be replaced. Due to its size (7.9 m diameter), the SDFIR is also difficult to install in remote locations. Based on the success of the Belfort double Alter wind shield, which is only 2 m in diameter, we hypothesized that a shield with a lowered porosity that is smaller than the SDFIR but larger than the Belfort double Alter would be as effective as the SDFIR. To validate this hypothesis, a smaller (4.9 m diameter) LPDF wind shield was designed and tested for use in the USCRN. The LPDF wind shield is described here, along with the results of a field experiment designed to evaluate the LPDF at three separate sites.

## 2 Methods

### 2.1 Shield design

In addition to decreasing the size of the wind shield, additional goals for the new shield design included using more durable materials, reducing the amount of labour required to construct the shield, and allowing easier access to the precipitation gauge within the shield. Improving gauge access was a priority in part because each USCRN site is typically visited only annually for routine maintenance, so volunteer site hosts are occasionally called upon to partially drain the 600- or 1000- mm capacity Geonor weighing gauges (Model T-200B-3, Geonor, Norway) employed by the network.

The new LPDF is constructed using chain link gate panels, which are widely available throughout the US. Each panel is 1.83 m wide and 1.22 m tall. Eight panels are used to form the outer octagonal shield, and four panels are used to create a concentric square inner shield. Unlike the SDFIR and the DFIR, which have a single Alter shield within the two outer wooden fences, the LPDF is comprised of only two shields. A comparison of the relative sizes of the outer shields of the DFIR, SDFIR, and LPDF is shown in Fig. 1. Vinyl slats (EZ slats®, Just Slats Co.) typically used to provide privacy and/or wind protection are installed within the chain link fence panels, providing a porosity of ~25%. The panels are mounted on galvanized poles using clamps designed for chain link fencing and gates, and one panel on both the inner and outer shields is hinged to allow easy access to the gauge. The height of the top of the inner shield is 0.20 m above the gauge inlet, and the top of the outer shield is 0.40 m above the gauge inlet. Shield heights are determined with respect to the top of the precipitation gauge inlet, because gauge inlet heights can vary depending on the maximum snow depth and the prevalence of drifting snow at a site. The design of the LPDF also allows it to be raised much more easily than a DFIR or SDFIR; raising a shield can be necessary when its initial installation height is too low, allowing drifting snow to accumulate within and around the shield.





## 2.2 Site selection

In addition to evaluating the LPDF at the NCAR/FAA/NOAA precipitation testbed in Marshall, CO (Rasmussen et
al., 2012; Baghapour et al., 2017), two USCRN sites with SDFIRs were chosen to test the LPDF. These USCRN sites were
selected for their potentially high wind speeds and frequent solid precipitation. Using 8 years (2008-2016) of daily
precipitation, air temperature, and wind speed data recorded from USCRN stations, all 114 conterminous USCRN sites with
SDFIRs were evaluated for potential inclusion in this study. 'Snow days' were defined as days with a total precipitation greater
than 1 mm and mean air temperature less than – 2℃. For every site, the total annual amount of solid precipitation recorded
during snow days was calculated, along with the number of snow days per year. Wind speeds during the snow days were also
evaluated; the mean wind speed for the snow days was calculated. In addition, the number of days with mean wind speeds
greater than 3.5 m s$^{-1}$ was determined. Based on the daily statistics, the wind speed distribution of the snow days was also
plotted for 10 of the snowiest and windiest USCRN sites (e.g. Fig. 2). The Boulder, CO, USCRN site (40.0353° N, -105.5407°
W) was chosen to test the LPDF, as it was clearly the windiest and snowiest site in the network. The Chatham, MI site
(46.3346°N, 86.9199°W) was selected as the second USCRN site due to the prevalence of all phases of precipitation, relatively
high wind speeds during snowfall, and the opportunity to test the LPDF in a different, non-alpine climate. All three sites are
shown in Fig. 3.

## 2.3 Installation

An LPDF was installed at the Marshall, CO testbed on 01/11/2018. This site was unique among the three LPDF
evaluation sites because in addition a SDFIR shield, it also included a DFIR. The LPDF at the Boulder, CO USCRN site was
installed on 28/11/2018, with the gauge inlet at a height of 1.77 m. The LPDF at the Chatham, MI site was installed on
23/06/2019 at a height of 2.13 m. Each of the two USCRN sites had pre-existing SDFIR shields. All of the shields included in
this evaluation contained heated Geonor weighing gauges (Geonor T-200B-3 All-weather precipitation gauge), with one
exception; at the Marshall, CO site, there were several periods when the Geonor gauge within the DFIR malfunctioned, and
measurements from an OTT-Pluvio$^2$ within a separate DFIR shield were used instead. These two precipitation gauge models
were previously shown to be interchangeable, and they were both used as references throughout WMO-SPICE (Nitu et al.,
2019). All of the Geonors were 600 mm capacity gauges, with the exception of the gauge within the Chatham, MI SDFIR,
which had a 1000 mm capacity. All of the Geonors also each had three vibrating wires. All of the precipitation gauge orifices
were heated, with the heaters activated only when the inlet temperature and the air temperature were both less than 2 ℃.

Among the pre-existing meteorological measurements available at the Marshall testbed, the sensors included in the
present evaluation included three fan-aspirated (Met-One Instruments, 076B Fan Aspirated Radiation Shield) air temperature
measurements (Thermometrics Corporation, PT1000 Platinum Resistance Thermometer) at a height of 1.5 m, and wind speed
measurements (RM Young Model 05103 Wind Monitor) at a height of 3 m and 10 m. The Chatham and Boulder USCRN sites
included the same fan-aspirated triplicate air temperature measurements as the Marshall testbed, which were also installed at





a height of 1.5 m. The USCRN sites also included a cup anemometer wind speed measurement (Met-One Model 014A) at a height of 1.5 m. For a more in-depth description of the standard suite of USCRN measurements see Diamond et al. (2013). In addition to the LPDF, Geonor weighing gauge, datalogger, and communications, an additional wind speed sensor (RM Young Model 05103 Wind Monitor) was installed at a height of 3.25 m at both USCRN sites for this intercomparison. All of the sites also included a precipitation detector (Vaisala Rain Detector, DRD11A), which was used to help identify periods when
precipitation occurred.

The Marshall measurements were recorded every minute, and the Chatham and Boulder measurements were recorded every 5 min. All of the measurements were transferred and archived in near-real-time. The Marshall field evaluation concluded on 26/09/2021, the Chatham field evaluation concluded on 19/12/2021, and the Boulder field evaluation concluded on 20/08/2021.

**2.4 Data analysis**

**2.4.1 Multi-seasonal and hourly precipitation**

Due to the effects of gauge uncertainty and small-scale spatial variability in precipitation, comparisons of hourly and even daily precipitation measurements recorded at the same site are subject to significant and seemingly random differences (Nitu et al., 2019). When comparing identical precipitation measurement configurations, these differences are not typically
associated with significant biases. Uncertainties in 30- or 60- min measurements make it more difficult to identify and quantify the biases associated with different types of wind shielding, particularly for measurements of solid precipitation, most of which are associated with low precipitation rates ($< 0.5$ mm hr$^{-1}$) (Kochendorfer et al., 2017b). For hydrology research, seasonal accumulations of solid precipitation are used to estimate the snow water equivalent of snow on the ground and to predict runoff and streamflow (Fekete et al., 2004; Boudhar et al., 2009). Because of this, longer term seasonal- or annual- scale
accumulations can be preferable for the comparison of different precipitation measurements and adjustments (e.g. Smith et al., 2019), and have even been used to develop and optimize precipitation gauge transfer functions (Kochendorfer et al., 2020). The comparison of long-term accumulations is in many ways a more demanding and representative test of different precipitation measurement configurations than the comparison of hourly or daily precipitation accumulations. For all of these reasons, time series of precipitation measurements accumulated over the entire length of the field campaign were central to the
LPDF evaluations.

Long-term season and annual precipitation accumulations were derived from the available gauge depths. The long-term precipitation accumulation time series must be developed with care, as the effects of evaporation, gauge maintenance, and missing data must be identified and treated appropriately. Typically, the three separate Geonor gauge depths recorded in each gauge were averaged together, but in some cases an individual noisy Geonor vibrating wire would be excluded from the
average. During periods when the OTT-Pluvio$^2$ data were used, only one precipitation gauge depth measurement was available. The 1-minute (Marshall) or 5-minute (Boulder and Chatham) gauge depth measurements were then examined to identify


unrealistically large changes. In addition, based on the precipitation detector measurements, changes in gauge depth that did not coincide with precipitation were discarded. At the Marshall testbed several measurement gaps occurred due to a loss of communications between the dataloggers and NCAR. During these periods the gauges continued to function and accumulate

precipitation, but without outputting their data in real time. These data were processed carefully so that the accumulated precipitation that occurred when the gauge data were not recorded was included in the long-term accumulations. This was possible in part because all of the gauges were serviced with oil, so it was not necessary to identify periods when evaporation was occurring. The long-term precipitation accumulations were used to evaluate the total accumulation and the seasonal course of precipitation accumulated within the different shield configurations. These long-term accumulations included all phases of

precipitation.

In addition, hourly precipitation was calculated as the hourly change in the gauge depth. The mean hourly air temperature was used to estimate the precipitation phase; every hourly precipitation measurement was classified as solid ($T_{air}$ < -2 ℃), mixed (2 ℃ ≥ $T_{air}$ ≥ -2 ℃), or liquid ($T_{air}$ > 2 ℃) based on $T_{air}$ (Wolff et al., 2015; Kochendorfer et al., 2017b). The hourly data were used to estimate separate phase-specific long-term accumulations for solid, mixed, and liquid precipitation.

The solid and mixed hourly precipitation measurements were also used to evaluate the LPDF catch efficiency (CE), which was the ratio of the amount of precipitation recorded by the LPDF-shielded gauge to the amount of precipitation recorded by the SDFIR-shielded gauge. For the evaluation of CE, when mean hourly wind speed was unavailable (or equal to 0.0 m s$^{-1}$) the hourly precipitation values were discarded. In addition, hourly precipitation values less than 0.25 mm were discarded; when either the LPDF-shielded gauge or the SDFIR-shielded gauge measured less than 0.25 mm in an hour, the

entire hour was excluded from the CE analysis. This was done mainly due to the well-documented increases in CE uncertainty for small values of precipitation (e.g. Kochendorfer et al., 2017b; Nitu et al., 2019).

### 2.4.2 Wind speed

At the Marshall testbed site, the gauge height wind speed ($U_{gh}$) was estimated using the 2 m height wind speed. During periods when it was equal to 0.0 m s$^{-1}$ the 3 m height wind speed was used, with the 3 m height wind speed divided by 1.09 to

approximate $U_{gh}$; the value of 1.09 was determined by comparing the available 3 m and 2 m wind speeds during precipitation. At the two CRN stations, the cup anemometer measurements recorded at a height of 1.5 m were used to estimate $U_{gh}$ based on a logarithmic vertical wind profile, the inlet heights, and the relationship between the 1.5 and 3.05 m high anemometer measurements (e.g. Kochendorfer et al., 2017a; Thom, 1975). Using this approach, the wind speed at the inlet height was estimated to be equal the 1.5 m height wind speed multiplied by 1.01 and 1.08 at the Boulder and Chatham sites, respectively;

these values differed from each other due to the exposure of the two sites and the installation height of the LPDF-shielded Geonors (1.77 m at Boulder and 2.13 m at Chatham). Average wind speeds were recorded every 1 min (Marshall) or 5 min (Boulder and Chatham) and averaged in 1 h intervals to correspond with the precipitation measurements.


### 2.4.3 Blowing snow

The effects of blowing snow were apparent in the preliminary *CE* evaluations, with *CE* values becoming unpredictable
above gauge height wind speeds of 9 or 10 m s$^{-1}$. Many factors affect the threshold wind speed above which snow on the
ground breaks loose, initiating saltation and lofting. Among them are the liquid water content of the snow and the age of the
snowpack (Schmidt, 1980, 1982). Values of the 10 m height wind speed threshold therefore vary widely, ranging from 4 – 14
m s$^{-1}$ on the Canadian prairies, but for dry snow they average about 8 m s$^{-1}$ (Li and Pomeroy, 1997). For the present work, to
conserve as many of the available measurements as possible, hourly precipitation measurements with $U_{gh} > 9$ m s$^{-1}$ were
excluded from the comparisons of the different precipitation gauge configurations.

## 3 Results and Discussion

### 3.1 Marshall, CO Precipitation Testbed

At the Marshall, CO precipitation testbed, the LPDF- and the SDFIR- shielded measurements compared well to each
other. The time series of the LPDF and SDFIR accumulations were almost indistinguishable from each other (Fig. 4). Over
the entire 2.9 years long intercomparison, the total LPDF precipitation differed by only 1% (-12 mm) of the total SDFIR
precipitation. As expected, the DFIR-shielded gauge (1309 mm) accumulated a little more than the SDFIR- (1265 mm) and
the LPDF- (1253 mm) shielded gauges.

As described in the methods section, hourly precipitation accumulations were also classified as solid, mixed, or liquid,
and accumulated separately over the course of the field experiment. The Marshall DFIR measurements were excluded from
these phase-specific accumulations because they were not always available at the same time as the other two gauges, and their
inclusion compromised the LPDF evaluation by diminishing the number of hours of precipitation available for comparison to
the SDFIR; because the LPDF is under evaluation as a replacement for the SDFIR within the USCRN, the comparison to the
SDFIR was prioritized over the comparison to the DFIR. The time series of the solid and mixed precipitation accumulations
demonstrated excellent agreement between the LPDF- and the SDFIR- shielded gauges (Figs. 5a and 5b). As expected, the
liquid precipitation measurements also compared well to each other (Fig. 5c).

The total of the phase-discriminated accumulations (Fig. 5) was less than the total of all the precipitation shown in
Fig. 4. The phase-discriminated measurements were subject to the additional requirement that the LPDF- and the SDFIR-
gauges were recording simultaneously, and communication lapses at the Marshall site caused data losses in the hourly phase-
discriminated measurements that did not affect the total accumulations.

### 3.2 Chatham, MI USCRN Site

Subject to 2600 mm of precipitation, the LPDF- and SDFIR- shielded precipitation measurements at the Chatham,
MI site compared quite well to each other. The total accumulations differed by only 1.01% (29 mm), and they tracked each





other closely throughout the measurement campaign (Fig. 5). When separated by precipitation type, the solid, mixed, and liquid precipitation accumulations also compared quite closely to each other (Fig. 6). Most of the precipitation at this site

occurred as rain, but it still experienced a significant amount of solid precipitation, and the total amounts of solid precipitation captured within the DFIR (348 mm) and the SDFIR (352 mm) were well within the margin of error for identical precipitation measurement configurations (Kochendorfer et al., 2017b).

### 3.3 Boulder, CO USCRN Site

As predicted from the site selection analysis, the Boulder, CO site experienced a significant amount of solid

precipitation. About 65% of the total accumulation of all precipitation types at the site was from solid precipitation. Despite these demanding conditions, the total accumulations from the SDFIR (1802 mm) and the LPDF (1786 mm) were within 1% of each other. The solid precipitation accumulation from the LPDF (1147 mm) was 1.4% less than the SDFIR accumulation (1164 mm) at this site, and this was mainly due to one event early in 2019 when the LPDF shield was clogged with wet snow, decreasing its porosity and potentially its performance. The time series of the precipitation accumulations from the LPDF

mimicked that of the SDFIR quite well (Figs. 7 and 8). The site did not experience a lot of mixed precipitation, but the LPDF measurements of mixed precipitation compared quite well to the corresponding SDFIR measurements (Fig. 8b).

### 3.4 Catch Efficiency

Measurements of the LPDF *CE* from all three sites were evaluated for their dependence on wind speed. The hourly *CE* measurements from all three sites were pooled together, and the mixed and solid precipitation measurements were

evaluated separately. Despite the well-documented effects of *CE* uncertainty (e.g. Hoover et al., 2021), the *CE* measurements for both solid and mixed precipitation were close to 1.0 even at high wind speeds, indicating that the *CE* of the LPDF does not decrease significantly with wind speed. Additionally, a t-test was performed to determine the probability that the differences between the hourly SDFIR- and LPDF- shielded solid precipitation measurements had a mean equal to zero. The t-test determined that there was no difference between the two datasets, with a 5% significance level and a 0.21% probability that

the results were incorrect.

### 4 Conclusions

A new LPDF wind shield has been designed for reference quality precipitation measurements. It is smaller than both the original DFIR, and also the SDFIR used in the USCRN. The LPDF is also constructed out of more durable materials than the wooden DFIR and SDFIR shields. For all of these reasons, the LPDF is more suitable for long-term and remote

measurements. Spanning over two years of measurements at three separate sites, the hourly LPDF- and SDFIR- shielded measurements of solid precipitation were statistically indistinguishable from each other, and the long-term precipitation





accumulations were remarkably similar to each other. Based on these analyses, the LPDF shield performed well, and can be used to replace the SDFIR without any introducing any new biases or inhomogeneities.

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





.


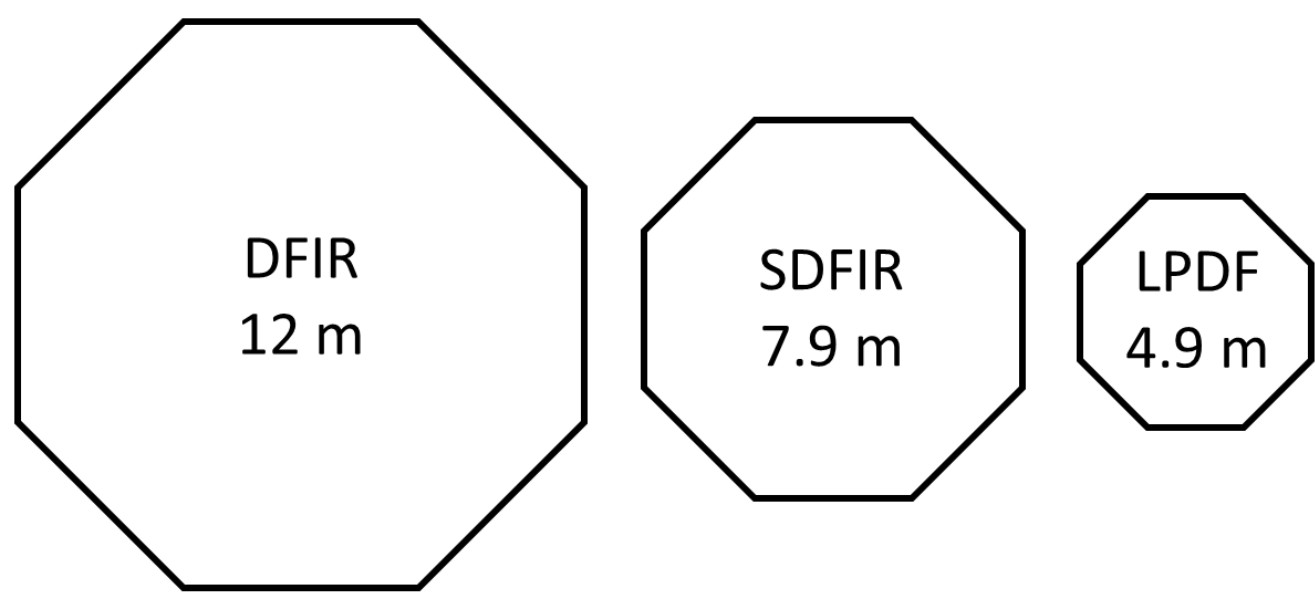

**Fig. 1: Illustration describing the relative sizes of different wind shields.**

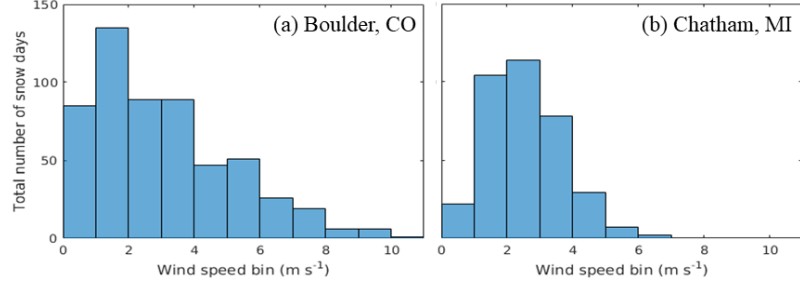

**Fig. 2: Wind speed distribution of snow days at the Boulder, CO (a), and Chatham, MI (b) USCRN sites.**


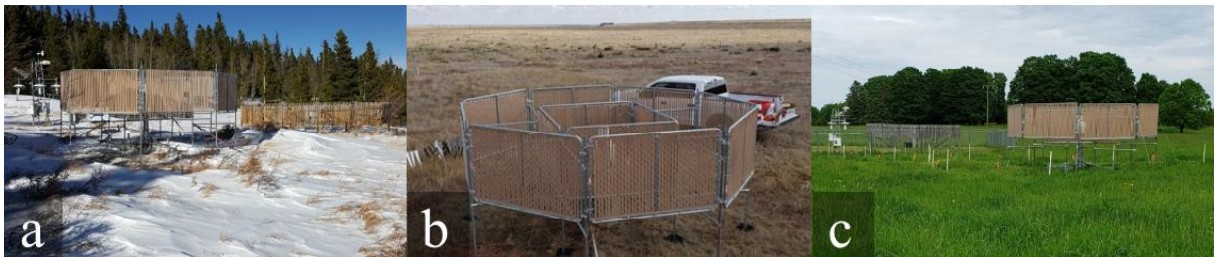

**Fig. 3: Photos of Low Porosity Double Fence (LPDF) shields installed at the Boulder, CO (a), Marshall, CO (b), and Chatham, MI (c) sites.**






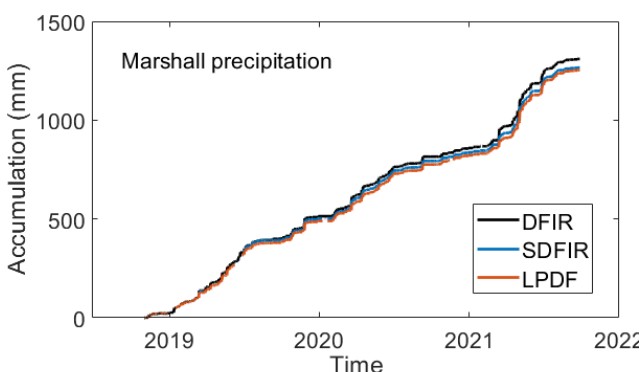

Fig. 4. Accumulation of all precipitation from the DFIR, SDFIR, and LPDF at the Marshall, CO precipitation testbed

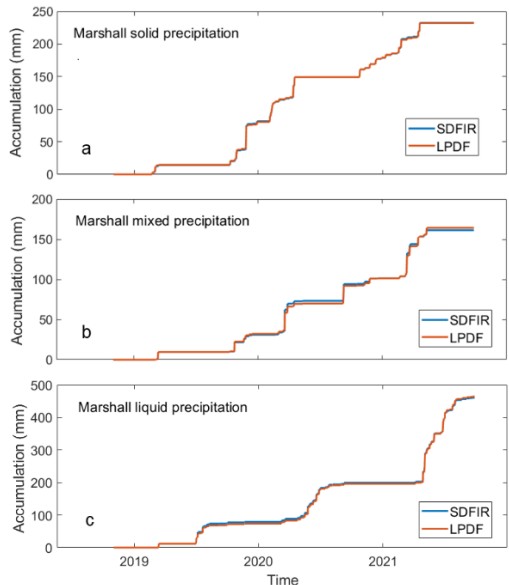

Fig. 5. Accumulated solid (a), mixed (b), and liquid (c) precipitation from the SDFIR and LPDF at the Marshall, CO precipitation.





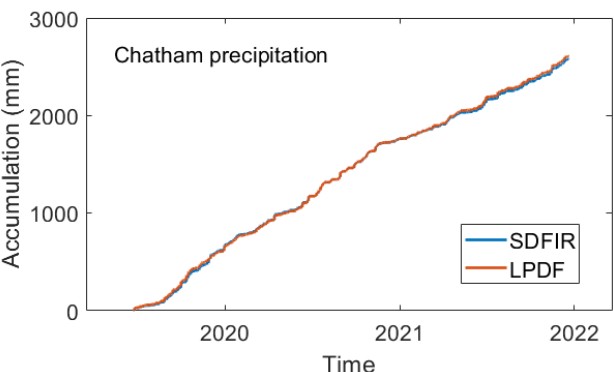


**Fig. 6. Accumulation of all precipitation from the DFIR, SDFIR, and LPDF at the Chatham, MI USCRN site.**

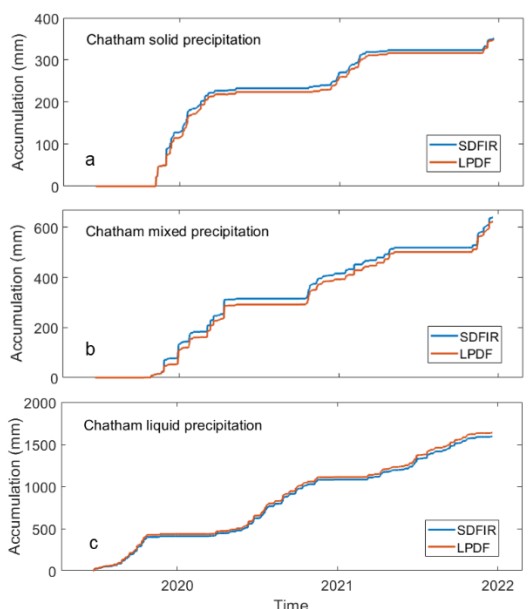

**Fig. 7. Accumulated solid (a), mixed (b), and liquid (c) precipitation from the SDFIR and LPDF at the Chatham, MI USCRN site.**





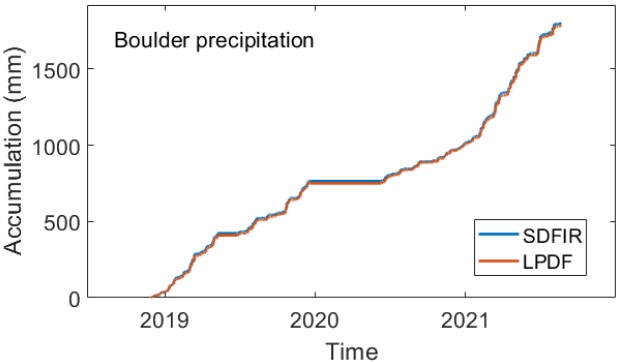

**Fig. 8. Accumulation of all precipitation from the DFIR, SDFIR, and LPDF at the Boulder, CO USCRN site.**

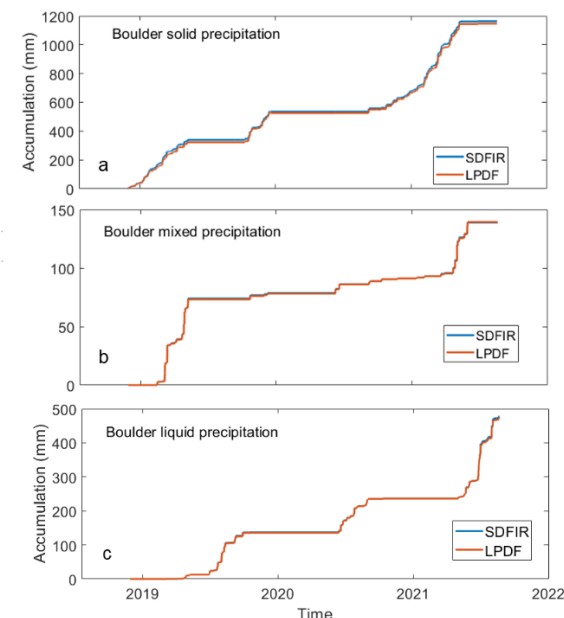

**Fig. 9. Accumulated solid (a), mixed (b), and liquid (c) precipitation from the SDFIR and LPDF at the Boulder, CO USCRN site.**





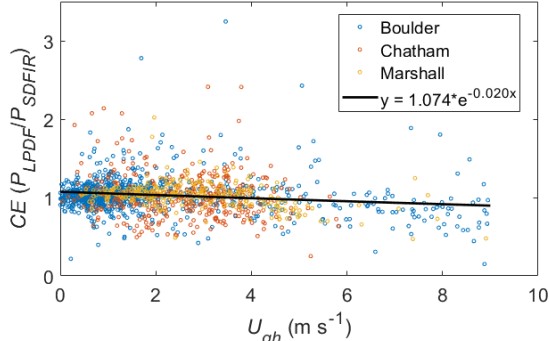

**Fig. 10. Solid precipitation catch efficiency (*CE*) plotted against the gauge height wind speed (*U<sub>gh</sub>*).**

Fig. 10. Solid precipitation catch efficiency ($CE$) plotted against the gauge height wind speed ($U_{gh}$).

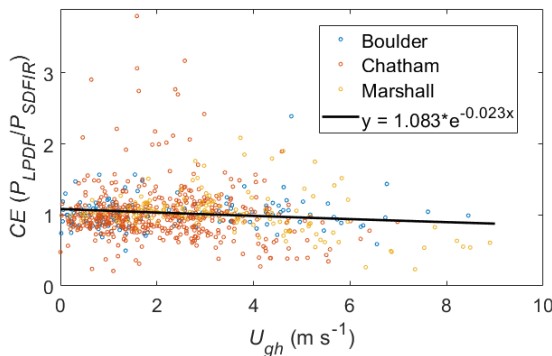

**Fig. 11. Mixed precipitation catch efficiency (*CE*) plotted against the gauge height wind speed (*U<sub>gh</sub>*).**

Fig. 11. Mixed precipitation catch efficiency ($CE$) plotted against the gauge height wind speed ($U_{gh}$).

### Author contribution

J. Kochendorfer helped lead the intercomparison, prepared the results, and wrote most of the manuscript. He also helped design the LPDF. T. Meyers helped discover the effects of porosity on wind shield efficacy, helped conceive of and design the LPDF, and helped complete the manuscript. M. Hall built the first LPDF, helped conceive of and design the LPDF, and coordinated the construction of the LPDFs and the maintenance of the measurements at the two USCRN sites. S. Landolt was responsible for the measurements at the Marshall site and for the installation of many of the wind shields and sensors at Marshall site. He

also contributed to the writing of the manuscript. H. Diamond provided guidance, support, and insight throughout the project. He also contributed to the writing of the manuscript.

### Competing interests

The authors declare that they have no conflict of interest.





**Acknowledgements**

A. Sczepanski from the Univ. of North Dakota provided guidance on the relationship between blowing snow and wind speed.