# Peer review of "A new reference-quality precipitation gauge wind shield"

_Atmospheric Measurement Techniques, 2023_

## Author Comment (AC1)

**Response to Referee #1:**

Congratulations for your article is very interesting and well treated. The paper is an important example of using alternatives to solid precipitation measurements in operational networks taking into account cost-value decisions. It is well suited to the audience of the journal and worth being published.

I like to remark the good experimental design and the strong methodology with 3 different sites for intercomparison. In my opinion, this is very important because it shows that replication in other sites is possible. The results are clear and the quality control of data is consistent with other similar studies

- Thank you.

However, some details and explanations are required in order to reproduce the LPDF in other sites, either for other intercomparisons or just for operational measurements.

I would suggest to include a more detailed description of the LPDF including a complete diagram (i.e cross section) and some details how the slats are installed within the chain link fence panels (before/after), how to avoid displacement of LPDF under strong winds, approximate dimensions of concrete footing and how to anchor and elevate the LPDF etc

- Some of the finer details of the engineering and components of the LPDF are still being worked out; the bracing, supports, mounting, and anchoring of the shields will be evaluated and potentially improved for longer-term installation, but we can provide more information and a drawing of the shield in the revised manuscript. Here is a rough draft, if we can't find a good spot for it in the manuscript itself, we will just provide a summary, and a more complete description in a supplement to the manuscript:

  The fence panels themselves are a standard gate panel that is available from construction supply and hardware outlets. The Supplement (to be provided with the revised manuscript) will include a list of the components of the LPDF, including the standard chain link hardware used to mount the fixed panels, and the two hinged openable panels (one on the interior fence, and one on the exterior fence). The clamps used to attach the panels to the supporting poles should be reinforced using metal screws, to keep the clamps in place and prevent movement in high winds.

  The slats were installed per the manufacturer's instructions, and easily slid and locked into place. They are vinyl and designed to be outdoors indefinitely, so weathering is not anticipated to be a significant problem. Furthermore, the slats can easily be replaced easily if they are damaged or subject to weathering over time.

  No concrete was used in the construction of the LPDF - the shield was designed to sit on top of the soil surface. Each pole supporting the panels sits on a reinforced 30 cm x 30 cm section of Geoblock. This is in part to ease installation. It also minimizes the effects

of frost heave. After the LPDF is assembled and braced, it is rigid, and is held in place by short guy wires anchored in the ground. The anchoring can be modified based on the soils at the site.

The panels are attached to the poles by clamps that can be loosened, allowing the height of the fence to be adjusted by sliding the loosened clamps up or down the poles. Once the desired height is reached, the clamps are tightened and additionally screwed in place."

Another important point for discussion is to analyze if given the objective of 25% of porosity the design can/could be slightly different (i.e panel dimensions, slats width, etc) obtaining similar results

- This is a good point! In the manuscript, we will acknowledge that more research may reveal alternative designs that would result in similar (or better) results that meet (or improve upon) design constraints. We did not have the resources to test different variants of the shield; testing many variants of the shield over the course of several years at three different sites would be a significant undertaking. Instead, we designed and tested a shield that met our design criteria, and was easy to find materials for in the US. A full examination of all of the possible effects of porosity, slat width, size, and shield height (with respect to the gauge inlet) was beyond the scope of this manuscript. But the authors agree that this is a worthwhile subject of discussion and future research. For several years prior to designing and testing the LPDF, we tried (and failed) to get support for numeric experiments focussed on testing the efficacy of different shield designs and shield heights, but we never succeeded in acquiring this support.

Another important point is a more detailed discussion about durability of LPDF compared with SDFIR and the approximate price different on material and time for installation

- We agree that this is another important point, especially as cost and ease-of-use is one of the main reasons for this research. In the revised manuscript, we will provide an estimate of the current costs of both the LPDF and the SDFIR.

  Regarding durability, the LPDF is assembled out of chain link fence material, thus it is designed by fence manufacturers to exist outdoors for an extended period of time. The SDFIR is made of pressure treated wood, which weathers over time. Some of the SDFIRs in the USCRN are 20 years old, but many require repair and replacement (of slats as well as supporting members) on a regular basis. Additionally, there are several fences in the USCRN that need to be completely replaced. In addition to the cost of building a new SDFIR, the replacement of an entire fence involves the disposal of close to 1000 kg of wood, which is not a trivial undertaking. We will summarize this in the manuscript.

Also some minor comments are provided below that I would ask the authors to consider before the paper can be accepted for publication.

1) Remove DFIR on the caption for figure 6 y 8

   - Good catch! Thank you. We will remove the caption.

2) Percentage of cases Ugh> 9 m/s

   - We will include that in the manuscript. This is also addressed in more detail in the response to Referee #2.

---

## Author Comment (AC2)

**Response to Referee #2:**

In this manuscript Kochendorfer et al. present a new type of precipitation gauge wind shield to mitigate undercatch of solid precipitation in windy conditions. The study describes the Low Porosity Double Fence (LPDF) which is smaller, more durable and easier to install and maintain in remote locations then the current reference-quality wind shields (such as the DFIR and the SDFIR) ((small) Double Fence Intercomparison Reference). I believe the work therefore presents a substantial new method and would recommend the publication of this manuscript after some minor revisions.

- Thank you.

 Below some more detailed remarks:

Page 1, lines 25-25: "This new wind shield is much smaller and easier to install and maintain" – is it also cheaper? In many organisations (and increasingly so) budget constraints can be an important factor.

- It is true that cost is important; the cost of replacement and maintenance within the USCRN was the primary motivator for this work. In response to Reviewer #1, we will estimate the cost difference between the two shields, including the amount of labor required for installation, and we will add this to the manuscript.

Page 2, line 64: perhaps the authors could briefly define/explain the meaning of porosity in the context of windshields, and how it is estimated/calculated.

- Porosity is defined as the amount of the surface area that is open, allowing air to pass through, divided by the total amount of surface area. It was calculated as the amount of fence panel surface area that was open (i.e. not blocked by slats and wire), divided by the total surface area of the fence panel. We will include this in the manuscript.

Page 3, line 95-97: "the design of the LPDF also allows it to be raised much more easily than the DFIR or SDFIR" – here and in general, it would be nice to include a technical drawing of the LPDF. This would also improve reproducibility of the work.

- A technical drawing will be added to the manuscript.

Page 4, site selection. Regarding the site selection procedure, how representative are the chosen sites for USCRN sites overall? I'm also missing a map with the site locations as well as wind statistics for the Marshall site in Figure 1. (consider having a,b and c panels in Figures 1 and 2 to be referring to the same sites).

- We will add a map showing the sites of the locations.

As described in the manuscript, we chose the Boulder and Chatham sites from among all of the USCRN sites because they experienced high winds during snow events. The goal was to test the LPDF at the most challenging sites in the USCRN, not the most representative. So, the sites are representative of the worst-case for the measurement of solid precipitation. Any differences found between the LPDF- and the SDFIR- shielded measurements at the chosen sites would presumably be much larger than found at representative sites. This will be explained in the manuscript.

Due to the design philosophy of the USCRN, even sites in regions that experience little or no solid precipitation are shielded by SDFIRs. Testing the LPDF at such a site, for example, would teach us very little about the performance of the LPDF, because undercatch due to wind is much larger for solid precipitation than for liquid precipitation.

Page 5, lines 142-155: In this paragraph the authors claim hourly and daily precipitation measurements are subject to random errors and therefore not appropriate for intercomparison studies. This seems in contradiction with paragraph 3.4 (and figures 10 and 11) where hourly catch efficiency measurements are shown. It also seems at odds with the introduction (page 2, lines 33-35) where several short term consequences of precipitation are mentioned. For avalanche predictions for example, accurate precipitation estimates at the precipitation event level would be more useful. Have the authors considered comparisons at the (long) event scale for example?

- The manuscript does not state that hourly measurements are inappropriate for intercomparison studies. Lines 142 - 155 describes the challenges and shortcomings of using hourly measurements for this type of work, but hourly measurements have been and will continue to be used for intercomparison studies. Hourly measurements are useful, but they have limitations, and seasonal accumulations have some advantages for comparing precipitation measurement systems and identifying precipitation biases.

  Figures 10 and 11 actually serve as good examples of the uncertainties and shortcomings of using hourly measurements; the significant uncertainty shown in those hourly CE measurements demonstrates the issue discussed in ln 142 - 155, rather than contradicts it.

  The WMO Solid Precipitation Intercomparison (Nitu et al., 2019) included an evaluation of the use of precipitation events versus 30-min, 60-min, and daily precipitation measurements. The conclusion was that due to the need to create a somewhat arbitrary definition of 'event', the wide range of event sizes (in mm) and lengths (in time), and the limited number of events in a given time period, hourly data was preferable for the calculation and evaluation of catch efficiencies. In more recent publications, the use of time series of seasonal (or shorter) accumulations has become more widely accepted for the comparison of different precipitation measurement configurations and the evaluation of transfer functions (e.g. Smith et al., 2020, Buisan et al., 2020, Pierre et al. 2019).

Page 5, line 152: why is a comparison of long-term accumulation more demanding? Surely on the contrary, it has the effect of averaging out small errors?

- Small errors are indeed averaged out within long-term accumulations, and that is actually why long-term accumulations are in some ways a better test of shielding. Small biases become more pronounced and easier to detect within long-term accumulations. Precipitation shielding differences often exhibit themselves more readily as biases, rather than small errors, as all solid precipitation measurement comparisons (including those of identical configurations) are subject to significant uncertainty over shorter time periods (and smaller amounts of precipitation). Additionally, when calculating catch efficiency to look at the sensitivity of the measurements to wind speed, many representative measurements must be excluded, so the measurements used to create long-term accumulations are arguably more representative of operational networks. This has been described in detail for tipping bucket gauge evaluations, much of which is also true (albeit to a lesser extent) for weighing gauges (Kochendorfer et al., 2020).

  The authors do agree however, that solid precipitation data is used for different applications, and precipitation measurement systems should be evaluated with this in mind. This is why both hourly accumulations and time series over longer accumulation times were both included in the present evaluation.

Page 6, line 167: "serviced with oil" might be a confusing term here. Perhaps a brief description in the methods section on the anti-freeze (if used) and oil layer - and why these are used, could be helpful to readers unfamiliar with weighing gauges.

- Thank you. We will clarify what is meant by "serviced with oil."

Page 6, line 178: could the authors comment on how prevalent situations were in which the SDFIR accumulated more than 0.25 mm in an hour and LPDF less than 0.25 mm, and vice versa? Was this negligible?

- Unfortunately, such situations were not negligible (Table R1). Solid precipitation rates are often quite low, and many measurements occur at or near 0.25 mm $hr^{-1}$ (e.g. Kochendorfer et al., 2017, Fig. 3). Additionally, at or near this rate of 0.25 mm $hr^{-1}$, the hourly measurements are subject to a significant amount of error/noise, because they are close to the measurement resolution of the gauge (e.g. Kochendorfer et al., 2017, and Kochendorfer et al., 2018). Because of this, we caution against reading too much into the results shown in Table R1. In addition, periods when one of the gauges was not working also contributed to the number of hourly measurements above the 0.25 mm threshold for each measurement configuration.

| Site | $P_{Hr\_SDFIR}$ > 0.25 m | $P_{Hr\_LPDF}$ > 0.25 m | $P_{Hr\_Both}$ > 0.25 m |
|---|---|---|---|
| Boulder | 1115 | 1032 | 957 |
| Chatham | 398 | 450 | 354 |
| Marshall | 298 | 336 | 284 |

- Table R1 shows the site name, the number of hours with solid precipitation ($T_{air}$ < -2 deg C) greater than 0.25 mm as measured by the SDFIR ($P_{Hr\_SDFIR}$ > 0.25 m), by the LPDF ($P_{Hr\_LPDF}$ > 0.25 m), and both gauges ($P_{Hr\_Both}$ > 0.25 m)

Page 7, line 197-200: again, what percentage of total measurements did this represent? Based on Fig 2, windspeeds above 9 m/s must have been a very rare phenomenon.

- It is true that hourly precipitation measurements with winds above 9 m s$^{-1}$ were rare. However, Figure 2 is not a good indicator of the number of hours with $U_{gh}$ > 9 m s$^{-1}$, because it is based on daily data, rather than hourly data.

    At Chatham, none of the solid or mixed precipitation hours were above the 9 m s$^{-1}$ threshold. At Marshall, none of the solid precipitation hours were above the 9 m s$^{-1}$ threshold, and 1.9% of the hourly mixed precipitation values were above the threshold. At the Boulder site, 2.2% of the solid precipitation values and none of the mixed values were above the blowing snow $U_{gh}$ threshold.

Page 7-8 paragraph 3.2 and Figure 7: Single events seem to have most affected the difference between the accumulations of solid and mixed precipitation in Chatham. Have the authors looked into this or could they comment on this?

- We are not sure what the cause of this was, but it was not considered significant enough to affect the broad conclusions of the manuscript.

Minor

Page 3, line 1: In addition .. additional (repetition).

- Thank you. We will change line 82 to read, "In addition to decreasing the size of the wind shield, goals for the new shield included…"

Page 4, line 102: "conterminous" not sure what conterminous is supposed to mean in this context – or why it is important that USCRN sites are conterminous.

- We will delete the word, "conterminous."

Page 4, lines 105-108: " Wind speeds during the snow days .. USCRN sites (e.g. Fig. 2)." These sentences seem unnecessarily repetitive, please consider revising them.

- "Wind speeds during the snow days were also evaluated; the mean wind speed for the snow days was calculated" will be deleted.

Page 7, lines 216-217: consider adding causal link between these two sentences for better comprehension, i.e. "precipitation shown in Fig.4. This is because the phase discriminated measurements" etc.

- These lines will be rewritten as follows: "The total of the phase-discriminated accumulations (Fig. 5) was less than the total of all the precipitation shown in Fig. 4. This is because the phase-discriminated measurements were subject to the additional requirement that the LPDF- and the SDFIR- gauges were recording simultaneously…"

Page 8, lines 231-232: consider reporting SDFIR and LPDF values in the same order throughout the manuscript.

- Thank you for this! We will report SDFIR and LPDF values in the same order throughout the manuscript.

**References**

Buisán, ST, Smith, CD, Ross, A, et al. The potential for uncertainty in Numerical Weather Prediction model verification when using solid precipitation observations. Atmos Sci Lett. 2020; 21: 21:e976. https://doi.org/10.1002/asl.976

Kochendorfer, J., and Coauthors, 2020: Undercatch Adjustments for Tipping-Bucket Gauge Measurements of Solid Precipitation. J. Hydrometeor., 21, 1193–1205, https://doi.org/10.1175/JHM-D-19-0256.1.

Kochendorfer, J., Nitu, R., Wolff, M., Mekis, E., Rasmussen, R., Baker, B., Earle, M. E., Reverdin, A., Wong, K., Smith, C. D., Yang, D., Roulet, Y.-A., Meyers, T., Buisan, S., Isaksen, K., Brækkan, R., Landolt, S., and Jachcik, A.: Testing and development of transfer functions for weighing precipitation gauges in WMO-SPICE, Hydrol. Earth Syst. Sci., 22, 1437–1452, https://doi.org/10.5194/hess-22-1437-2018, 2018.

Kochendorfer, J., Rasmussen, R., Wolff, M., Baker, B., Hall, M. E., Meyers, T., Landolt, S., Jachcik, A., Isaksen, K., Brækkan, R., and Leeper, R.: The quantification and correction of wind-induced precipitation measurement errors, Hydrol. Earth Syst. Sci., 21, 1973–1989, https://doi.org/10.5194/hess-21-1973-2017, 2017.

Pierre, A., S. Jutras, C. Smith, J. Kochendorfer, V. Fortin, and F. Anctil, 2019: Evaluation of Catch Efficiency Transfer Functions for Unshielded and Single-Alter-Shielded Solid Precipitation Measurements. J. Atmos. Oceanic Technol., 36, 865–881, https://doi.org/10.1175/JTECH-D-18-0112.1.

Smith, C. D., Ross, A., Kochendorfer, J., Earle, M. E., Wolff, M., Buisán, S., Roulet, Y.-A., and Laine, T.: Evaluation of the WMO Solid Precipitation Intercomparison Experiment (SPICE) transfer functions for adjusting the wind bias in solid precipitation measurements, Hydrol. Earth Syst. Sci., 24, 4025–4043, https://doi.org/10.5194/hess-24-4025-2020, 2020.

---

## Author Response (AR1)

**Response to Referee #1:**

**Referee comment:** Congratulations for your article is very interesting and well treated. The paper is an important example of using alternatives to solid precipitation measurements in operational networks taking into account cost-value decisions. It is well suited to the audience of the journal and worth being published.

I like to remark the good experimental design and the strong methodology with 3 different sites for intercomparison. In my opinion, this is very important because it shows that replication in other sites is possible. The results are clear and the quality control of data is consistent with other similar studies

However, some details and explanations are required in order to reproduce the LPDF in other sites, either for other intercomparisons or just for operational measurements.

I would suggest to include a more detailed description of the LPDF including a complete diagram (i.e cross section) and some details how the slats are installed within the chain link fence panels (before/after), how to avoid displacement of LPDF under strong winds, approximate dimensions of concrete footing and how to anchor and elevate the LPDF etc

**Response:** Some of the finer details of the engineering and components of the LPDF are still being worked out; the bracing, supports, mounting, and anchoring of the shields will be evaluated and potentially improved for longer-term installation, but we have provided more information, a supplemental list of components, and a new drawing of the shield in the revised manuscript.

**Manuscript changes:**

Added to Section 2.1: A reference to Figure 1, and "Some of the construction details such as bracing are still under development, but a Supplement includes a list of the LPDF components, including the standard chain link hardware used to mount the fixed panels, and the two hinged openable panels (one on the interior fence, and one on the exterior fence). The clamps used to attach the panels to the supporting poles should be reinforced using metal screws, to keep the clamps in place and prevent movement in high winds…" "The slats were installed per the manufacturer's instructions, and easily slid and locked into place. They are vinyl and designed to be outdoors indefinitely, so weathering is not anticipated to be a significant problem. Furthermore, the slats can be replaced easily if they are damaged or subject to weathering over time…" "The panels are attached to the poles by clamps that can be loosened, allowing the height of the fence to be adjusted by sliding the loosened clamps up or down the poles. Once the desired height is reached, the clamps are tightened and additionally screwed in place. No concrete was used in the construction of the LPDF - the shield was designed to sit on top of the soil surface. Each pole supporting the panels sits on a reinforced 30 cm x 30 cm section of Geoblock. This is in part to ease installation. It also minimizes the effects of frost heave. After the LPDF is assembled and braced, it is rigid, and is held in place by short guy wires anchored

in the ground. Anchors can be concreted in place, or screw in anchors can be used, and the anchoring methods may require modification based on the ground structure at the site.

The cost of the materials to build the LPDF are low (~$2,000), but are higher than the cost of materials required to build the larger SDFIR (~$1,400). However, the amount of labor required to build the LPDF (8 h) is significantly less than is needed for the SDFIR (24 h). The cost of SDFIR maintenance (which varies considerably by site), eventual disposal (entailing ~900 kg of wood), and replacement must also be considered when comparing the use of both shields."

The panels are attached to the poles by clamps that can be loosened, allowing the height of the fence to be adjusted by sliding the loosened clamps up or down the poles.  Once the desired height is reached, the clamps are tightened and additionally screwed in place."

**Referee comment:** Another important point for discussion is to analyze if given the objective of 25% of porosity the design can/could be slightly different (i.e panel dimensions, slats width, etc) obtaining similar results

**Response:** This is a good point! In the manuscript, we will acknowledge that more research may reveal alternative designs that would result in similar (or better) results that meet (or improve upon) design constraints. We did not have the resources to test different variants of the shield; testing many variants of the shield over the course of several years at three different sites would be a significant undertaking. Instead, we designed and tested a shield that met our design criteria, and was easy to find materials for in the US. A full examination of all of the possible effects of porosity, slat width, size, and shield height (with respect to the gauge inlet) was beyond the scope of this manuscript. But the authors agree that this is a worthwhile subject of discussion and future research. For several years prior to designing and testing the LPDF, we tried (and failed) to get support for numeric experiments focussed on testing the efficacy of different shield designs and shield heights, but we never succeeded in acquiring this support.

**Manuscript changes:** The following paragraph was added to the Conclusions section, "Future research may reveal alternative designs that result in improved results while simultaneously meeting different design constraints. We did not have the resources to test different variants of the shield; testing many variants of the shield over the course of several years at three different sites would be a significant undertaking. Instead, we designed and tested a shield that met USCRN design criteria using materials that are widely available in the US. A full examination of the effects of porosity, slat width, shield size, and shield height (with respect to the gauge inlet) was beyond the scope of this manuscript. However, clearly the results of the evaluation of the LPDF and the Belfort double Alter indicate that this is a worthwhile subject of more in-depth study. Numerical modeling could also be used to aid in initial efforts to experiment with different wind shield designs."

**Referee comment:** Another important point is a more detailed discussion about durability of LPDF compared with SDFIR and the approximate price different on material and time for installation

**Response:** We agree that this is another important point, especially as cost and ease-of-use is one of the main reasons for this research. In the revised manuscript, we will provide an estimate of the current costs of both the LPDF and the SDFIR.

Regarding durability, the LPDF is assembled out of chain link fence material, thus it is designed by fence manufacturers to exist outdoors for an extended period of time. The SDFIR is made of pressure treated wood, which weathers over time. Some of the SDFIRs in the USCRN are 20 years old, but many require repair and replacement (of slats as well as supporting members) on a regular basis. Additionally, there are several fences in the USCRN that need to be completely replaced. In addition to the cost of building a new SDFIR, the replacement of an entire fence involves the disposal of close to 1000 kg of wood, which is not a trivial undertaking. We will summarize this in the manuscript.

**Manuscript changes:** The following paragraph has been added to the end of the Shield Design section: "The cost of the materials to build the LPDF are low (~$2,000), but are higher than the cost of materials required to build the larger SDFIR (~$1,400). However, the amount of labor required to build the LPDF (8 h) is significantly less than is needed for the SDFIR (24 h). The cost of SDFIR maintenance (which varies considerably by site), eventual disposal (entailing ~900 kg of wood), and replacement must also be considered when comparing the use of both shields.

**Referee comment:** Also some minor comments are provided below that I would ask the authors to consider before the paper can be accepted for publication.

1) Remove DFIR on the caption for figure 6 y 8

**Response:** Good catch! Thank you.

**Manuscript changes:** We will remove the caption.

**Referee comment:** 2) Percentage of cases Ugh> 9 m/s

**Response:** We will include that in the manuscript. This is also addressed in more detail in the response to Referee #2.

**Manuscript changes:**

**Response to Referee #2:**

**Referee comment:** In this manuscript Kochendorfer et al. present a new type of precipitation gauge wind shield to mitigate undercatch of solid precipitation in windy conditions. The study describes the Low Porosity Double Fence (LPDF) which is smaller, more durable and easier to

install and maintain in remote locations then the current reference-quality wind shields (such as the DFIR and the SDFIR) ((small) Double Fence Intercomparison Reference). I believe the work therefore presents a substantial new method and would recommend the publication of this manuscript after some minor revisions.

**Response:** Thank you.

**Referee comment:** Below some more detailed remarks:

Page 1, lines 25-25: "This new wind shield is much smaller and easier to install and maintain" – is it also cheaper? In many organisations (and increasingly so) budget constraints can be an important factor.

**Response:** It is true that cost is important; the cost of replacement and maintenance within the USCRN was the primary motivator for this work. In response to Reviewer #1, we estimated the cost difference between the two shields, including the amount of labor required for installation, and we added a summary of the cost comparison to the manuscript.

**Manuscript changes:** The cost of the materials to build the LPDF are low (~$2,000), but are higher than the cost of materials required to build the larger SDFIR (~$1,400). However, the amount of labor required to build the LPDF (8 h) is significantly less than is needed for the SDFIR (24 h). The cost of SDFIR maintenance (which varies considerably by site), eventual disposal (entailing ~900 kg of wood), and replacement must also be considered when comparing the use of both shields.

**Referee comment:** Page 2, line 64: perhaps the authors could briefly define/explain the meaning of porosity in the context of windshields, and how it is estimated/calculated.

**Response:** Porosity is defined as the amount of the surface area that is open, allowing air to pass through, divided by the total amount of surface area. It was calculated as the amount of fence panel surface area that was open (i.e. not blocked by slats and wire), divided by the total surface area of the fence panel. We will include this in the manuscript.

**Manuscript changes:** This has been added to the Introduction: "Porosity is defined as the amount of the surface area that is open, allowing air to pass through, divided by the total amount of surface area." And this has been added to the Shield Design Section, "…, with the porosity calculated as the amount of fence panel surface area that was open (i.e. not blocked by slats and wire), divided by the total surface area of the fence panel."

**Referee comment:** Page 3, line 95-97: "the design of the LPDF also allows it to be raised much more easily than the DFIR or SDFIR" – here and in general, it would be nice to include a technical drawing of the LPDF. This would also improve reproducibility of the work.

**Manuscript changes:** A technical drawing (Fig. 1) has been added to the manuscript.

**Referee comment:** Page 4, site selection. Regarding the site selection procedure, how representative are the chosen sites for USCRN sites overall? I'm also missing a map with the site locations as well as wind statistics for the Marshall site in Figure 1. (consider having a,b and c panels in Figures 1 and 2 to be referring to the same sites).

**Response:** As described in the manuscript, we chose the Boulder and Chatham sites from among all of the USCRN sites because they experienced high winds during snow events. The goal was to test the LPDF at the most challenging sites in the USCRN, not the most representative. So, the sites are representative of the worst-case for the measurement of solid precipitation. Any differences found between the LPDF- and the SDFIR- shielded measurements at the chosen sites would presumably be much larger than found at representative sites. This will be explained in the manuscript.

Due to the design philosophy of the USCRN, even sites in regions that experience little or no solid precipitation are shielded by SDFIRs. Testing the LPDF at such a site, for example, would teach us very little about the performance of the LPDF, because undercatch due to wind is much larger for solid precipitation than for liquid precipitation.

**Manuscript changes:** The following has been added to Section 2.2: "The goal was to select sites where precipitation gauge shielding was critical to accurate precipitation measurement. Because of this, differences found between the LPDF- and the SDFIR- shielded measurements at the selected USCRN sites would presumably be much larger than found at more representative sites."

We also added a map showing the sites of the locations (Fig. 4).

**Referee comment:** Page 5, lines 142-155: In this paragraph the authors claim hourly and daily precipitation measurements are subject to random errors and therefore not appropriate for intercomparison studies. This seems in contradiction with paragraph 3.4 (and figures 10 and 11) where hourly catch efficiency measurements are shown. It also seems at odds with the introduction (page 2, lines 33-35) where several short term consequences of precipitation are mentioned. For avalanche predictions for example, accurate precipitation estimates at the precipitation event level would be more useful. Have the authors considered comparisons at the (long) event scale for example?

**Response:** The manuscript does not state that hourly measurements are inappropriate for intercomparison studies. Lines 142 - 155 describes the challenges and shortcomings of using hourly measurements for this type of work, but hourly measurements have been and will continue to be used for intercomparison studies. Hourly measurements are useful, but they have limitations, and seasonal accumulations have some advantages for comparing precipitation measurement systems and identifying precipitation biases.

Figures 10 and 11 actually serve as good examples of the uncertainties and shortcomings of using hourly measurements; the significant uncertainty shown in those hourly CE measurements demonstrates the issue discussed in ln 142 - 155, rather than contradicts it.

The WMO Solid Precipitation Intercomparison (Nitu et al., 2019) included an evaluation of the use of precipitation events versus 30-min, 60-min, and daily precipitation measurements. The conclusion was that due to the need to create a somewhat arbitrary definition of 'event', the wide range of event sizes (in mm) and lengths (in time), and the limited number of events in a given time period, hourly data was preferable for the calculation and evaluation of catch efficiencies. In more recent publications, the use of time series of seasonal (or shorter) accumulations has become more widely accepted for the comparison of different precipitation measurement configurations and the evaluation of transfer functions (e.g. Smith et al., 2020, Buisan et al., 2020, Pierre et al. 2019).

**Referee comment:** Page 5, line 152: why is a comparison of long-term accumulation more demanding? Surely on the contrary, it has the effect of averaging out small errors?

**Response:** Small errors are indeed averaged out within long-term accumulations, and that is actually why long-term accumulations are in some ways a better test of shielding. Biases are more pronounced and easier to detect within long-term accumulations. Precipitation shielding differences often exhibit themselves more readily as biases, rather than small errors, as all solid precipitation measurement comparisons (including those of identical configurations) are subject to significant uncertainty over shorter time periods (and smaller amounts of precipitation). Additionally, when calculating catch efficiency to look at the sensitivity of the measurements to wind speed, many representative measurements must be excluded, so the measurements used to create long-term accumulations are arguably more representative of operational networks. This has been described in detail for tipping bucket gauge evaluations (Kochendorfer et al., 2020), much of which is also true (albeit to a lesser extent) for weighing gauges.

The authors do agree however, that solid precipitation data is used for different applications, and precipitation measurement systems should be evaluated with this in mind. This is why hourly accumulations and time series over longer accumulation times were both included in the present evaluation.

**Referee comment:** Page 6, line 167: "serviced with oil" might be a confusing term here. Perhaps a brief description in the methods section on the anti-freeze (if used) and oil layer - and why these are used, could be helpful to readers unfamiliar with weighing gauges.

**Response:** Thank you. We will clarify what is meant by "serviced with oil."

**Manuscript changes:** The sentence has been expanded as follows: "This was possible in part because all of the gauges had oil added to their collection buckets to minimize the evaporation of water and anti-freeze, so it was not necessary to identify and remove periods when evaporation was occurring."

**Referee comment:** Page 6, line 178: could the authors comment on how prevalent situations were in which the SDFIR accumulated more than 0.25 mm in an hour and LPDF less than 0.25 mm, and vice versa? Was this negligible?

**Response:** Unfortunately, such situations were not negligible (Table R1). Solid precipitation rates are often quite low, and many measurements occur at or near 0.25 mm hr$^{-1}$ (e.g. Kochendorfer et al., 2017, Fig. 3). Additionally, at or near this rate of 0.25 mm hr$^{-1}$, the hourly measurements are subject to a significant amount of error/noise, because they are close to the measurement resolution of the gauge (e.g. Kochendorfer et al., 2017, and Kochendorfer et al., 2018). Because of this, we caution against reading too much into the results shown in Table R1. In addition, periods when one of the gauges was not working also contributed to the number of hourly measurements above the 0.25 mm threshold for each measurement configuration.

| Site | $P_{Hr\_SDFIR} > 0.25$ m | $P_{Hr\_LPDF} > 0.25$ m | $P_{Hr\_Both} > 0.25$ m |
|------|------|------|------|
| Boulder | 1115 | 1032 | 957 |
| Chatham | 398 | 450 | 354 |
| Marshall | 298 | 336 | 284 |

Table R1 shows the site name, the number of hours with solid precipitation ($T_{air} < -2$ deg C) greater than 0.25 mm as measured by the SDFIR ($P_{Hr\_SDFIR} > 0.25$ m), by the LPDF ($P_{Hr\_LPDF} > 0.25$ m), and both gauges ($P_{Hr\_Both} > 0.25$ m)

**Referee comment:** Page 7, line 197-200: again, what percentage of total measurements did this represent? Based on Fig 2, windspeeds above 9 m/s must have been a very rare phenomenon.

**Response:** It is true that hourly precipitation measurements with gauge height winds ($U_{gh}$) above 9 m s$^{-1}$ were rare. However, Figure 2 is not a good indicator of the number of hours with $U_{gh} > 9$ m s$^{-1}$, because it is based on daily data, rather than hourly data.

At Chatham, none of the solid or mixed precipitation hours were above the 9 m s$^{-1}$ $U_{gh}$ threshold. At Marshall, none of the solid precipitation hours were above the 9 m s$^{-1}$ threshold, and 1.9% of the hourly mixed precipitation values were above the threshold. At the Boulder site, 2.2% of the solid precipitation values and none of the mixed values were above the blowing snow $U_{gh}$ threshold.

**Referee comment:** Page 7-8 paragraph 3.2 and Figure 7: Single events seem to have most affected the difference between the accumulations of solid and mixed precipitation in Chatham. Have the authors looked into this or could they comment on this?

**Response:** We are not sure what the cause of this was, but it was not considered significant enough to affect the broad conclusions of the manuscript.

**Minor Referee #2 Comments:**

**Referee comment:** Page 3, line 1: In addition .. additional (repetition).

**Response:** Thank you!

**Manuscript changes:** We changed line 82 to read, "In addition to decreasing the size of the wind shield, *other* goals for the new shield included…"

**Referee comment:** Page 4, line 102: "conterminous" not sure what conterminous is supposed to mean in this context – or why it is important that USCRN sites are conterminous.

**Manuscript changes:** We deleted the word, "conterminous."

**Referee comment:** Page 4, lines 105-108: "Wind speeds during the snow days .. USCRN sites (e.g. Fig. 2)." These sentences seem unnecessarily repetitive, please consider revising them.

**Response:** Thank you!

**Manuscript changes:** This sentence has been deleted:  "Wind speeds during the snow days were also evaluated; the mean wind speed for the snow days was calculated".

**Referee comment:** Page 7, lines 216-217: consider adding causal link between these two sentences for better comprehension, i.e. "precipitation shown in Fig.4. This is because the phase discriminated measurements" etc.

**Manuscript changes:** These lines have been rewritten as follows: "The total of the phase-discriminated accumulations (Fig. 5) was less than the total of all the precipitation shown in Fig. 4. This is because the phase-discriminated measurements were subject to the additional requirement that the LPDF- and the SDFIR- gauges were recording simultaneously…"

**Referee comment:** Page 8, lines 231-232: consider reporting SDFIR and LPDF values in the same order throughout the manuscript.

**Response:** Thank you for this!

**Manuscript changes:** Throughout the manuscript, the LPDF and SDFIR results are now reported in the same order.

**References**

Buisán, ST, Smith, CD, Ross, A, et al. The potential for uncertainty in Numerical Weather Prediction model verification when using solid precipitation observations. Atmos Sci Lett. 2020; 21: 21:e976. https://doi.org/10.1002/asl.976

Kochendorfer, J., and Coauthors, 2020: Undercatch Adjustments for Tipping-Bucket Gauge Measurements of Solid Precipitation. J. Hydrometeor., 21, 1193–1205, https://doi.org/10.1175/JHM-D-19-0256.1.

Kochendorfer, J., Nitu, R., Wolff, M., Mekis, E., Rasmussen, R., Baker, B., Earle, M. E., Reverdin, A., Wong, K., Smith, C. D., Yang, D., Roulet, Y.-A., Meyers, T., Buisan, S., Isaksen, K., Brækkan, R., Landolt, S., and Jachcik, A.: Testing and development of transfer functions for weighing precipitation gauges in WMO-SPICE, Hydrol. Earth Syst. Sci., 22, 1437–1452, https://doi.org/10.5194/hess-22-1437-2018, 2018.

Kochendorfer, J., Rasmussen, R., Wolff, M., Baker, B., Hall, M. E., Meyers, T., Landolt, S., Jachcik, A., Isaksen, K., Brækkan, R., and Leeper, R.: The quantification and correction of wind-induced precipitation measurement errors, Hydrol. Earth Syst. Sci., 21, 1973–1989, https://doi.org/10.5194/hess-21-1973-2017, 2017.

Pierre, A., S. Jutras, C. Smith, J. Kochendorfer, V. Fortin, and F. Anctil, 2019: Evaluation of Catch Efficiency Transfer Functions for Unshielded and Single-Alter-Shielded Solid Precipitation Measurements. J. Atmos. Oceanic Technol., 36, 865–881, https://doi.org/10.1175/JTECH-D-18-0112.1.

Smith, C. D., Ross, A., Kochendorfer, J., Earle, M. E., Wolff, M., Buisán, S., Roulet, Y.-A., and Laine, T.: Evaluation of the WMO Solid Precipitation Intercomparison Experiment (SPICE) transfer functions for adjusting the wind bias in solid precipitation measurements, Hydrol. Earth Syst. Sci., 24, 4025–4043, https://doi.org/10.5194/hess-24-4025-2020, 2020.

**Revised manuscript, with track changes (turn track changes off before copying):**